# Characterization of VvmiR166s-Target Modules and Their Interaction Pathways in Modulation of Gibberellic-Acid-Induced Grape Seedless Berries

**DOI:** 10.3390/ijms242216279

**Published:** 2023-11-14

**Authors:** Yunhe Bai, Zhuangwei Wang, Linjia Luo, Xuxian Xuan, Wei Tang, Ziyang Qu, Tianyu Dong, Ziyang Qi, Mucheng Yu, Weimin Wu, Jinggui Fang, Chen Wang

**Affiliations:** 1College of Horticulture, Nanjing Agricultural University, Nanjing 210095, China; 2020204032@stu.njau.edu.cn (Y.B.);; 2Jiangsu Academy of Agricultural Sciences, Institute of Pomology, Nanjing 210014, China; 20030021@jaas.ac.cn (Z.W.); 19900009@jaas.ac.cn (W.W.)

**Keywords:** seedless grape, gibberellin, VvmiR166s, HD-Zip III, lignin

## Abstract

Exogenous GA is widely used to efficiently induce grape seedless berry development for significantly improving berry quality. Recently, we found that VvmiR166s are important regulators of response to GA in grapes, but its roles in GA-induced seedless grape berry development remain elusive. Here, the precise sequences of VvmiR166s and their targets *VvREV*, *VvHB15* and *VvHOX32* were determined in grape cv. ‘*Rosario Bianco*’, and the cleavage interactions of VvmiR166s-*VvHB15/VvHOX32/VvREV* modules and the variations in their cleavage roles were confirmed in grape berries. Exogenous GA treatment significantly induced a change in their expression correlations from positive to negative between VvmiR166s and their target genes at the seeds during the stone-hardening stages (32 DAF–46 DAF) in grape berries, indicating exogenous GA change action modes of VvmiR166s on their targets in this process, in which exogenous GA mainly enhanced the negative regulatory roles of VvmiR166s on *VvHB15* among all three VvmiR166s-target pairs. The transient OE-VvmiR166a-h/OE-VvHB15 in tobacco confirmed that out of the VvmiR166 family, VvmiR166h/a/b might be the main factors in modulating lignin synthesis through inhibiting *VvHB15*, of which VvmiR166h-*VvHB15*-*NtPAL4/NtCCR1/NtCCR2/NtCCoAMT5/NtCOMT1* and VvmiR166a/b-*VvHB15*-*NtCAD1* are the potential key regulatory modules in lignin synthesis. Together with the GA-induced expression modes of VvmiR166s-*VvHB15* and genes related to lignin synthesis in grape berries, we revealed that GA might repress lignin synthesis mainly by repressing *VvCAD1/VvCCR2/VvPAL2/VvPAL3/Vv4CL/VvLac7* levels via mediating VvmiR166s-*VvHB15* modules in GA-induced grape seedless berries. Our findings present a novel insight into the roles of VvmiR66s that are responsive to GA in repressing the lignin synthesis of grape seedless berries, with different lignin-synthesis-enzyme-dependent action pathways in diverse plants, which have important implications for the molecular breeding of high-quality seedless grape berries.

## 1. Introduction

Grapevine (*Vitis vinifera* L.) is amongst the most important fruit crops cultivated worldwide for its berries with multiple excellent traits. Out of them, seedlessness is the economic and quality trait favored by consumers. Nowadays, exogenous gibberellin (GA) is widely used to produce seedless berries through inducing parthenocarpy in many grape cultivars like ‘*Wink*’, ‘*Zuijinxiang*’, ‘*Fujiminori*’, ‘*Kyoho*’ and ‘*Early Sweet*’ [1,2,3,4,5,6]. Moreover, the ovules and seed stones are drastically inhibited during GA-induced grape I have checked and revised all.seedless berry development, whereas until now, this research mainly focused on the inhibited ovule development or seed embryo abortion induced by exogenous GA [4,5,6], but the aspect of GA-repressed seed-stone development is poorly reported. This has inspired us to study how GA mediates the inhibition of seed stones during seedless berry formation.

GA is one of the most important phytohormones and has been reported to function in hypocotyl elongation, seed germination, leaf development and flower induction in a wide range of plant species [7,8,9,10]. In grape, GA plays a significant role in its flower development and seedless berry induction and expansion, which are critical traits for high-quality grape berries [3,11]. In recent years, studies have reported GA effects on lignin content. Exogenous GA application to some plants like *Pyrus pyrifolia* [12], *Myrica rubra* [13] and *Oryza sativa* [14] led to the decrease in lignin content, as well as expression changes of lignin synthesis genes. Our previous research found that more than 130 VvmiRNAs could respond to GA in grape berries [15], and the responsive mode to GA of different VvmiRNAs-targets depends on the *cis*-elements, including hormone response elements, in their respective promoters [3]. Of them, VvmiR159, VvmiR160, VvmiR166, VvmiR167, VvmiR171, VvmiR172, VvmiR397 and VvmiR3633a are important conserved VvmiRNA families responding to exogenous GA participating in the induced process of grape seedless berries [16,17]; especially, VvmiR397 and VvmiR166 might be involved in the modulation of lignin synthesis through mediating their target genes like *VvLACs* and *VvHBs* during seed-stone development in grape berries. Since VvmiR397 is the content for one of our works, we focused here on the regulatory roles of VvmiR166 in the GA-signal-mediated lignin synthesis of seed stones of grape seedless berries.

MiR166 is a highly conserved multi-gene family of miRNAs and about 262 miR166 members have been identified in 45 plant species according to the miRBase database (http://www.mirbase.org) (accessed on 15 September 2017), implying that miR166s might possess a high degree of functional redundancy [18,19]. However, spatiotemporal expression patterns of *miR166* genes suggested that functional specificity is also present in miR166 family members [20,21]. For instance, miR166a and miR166b were found to be specifically expressed during post-embryonic meristem and root endodermis development in *Arabidopsis* [22]. Likewise, miR166a, miR166b and miR166g were found to be highly expressed during early embryogenesis, whereas the expression level of other miR166 members was not detected in *Arabidopsis* [20]. The various spatiotemporal expression profiles of miR166 members indicate the diversity of their potential roles, and interpreting this diversification is an important step toward understanding their fine-tuned regulatory roles in plant development.

Previous studies have reported that miR166 targets five members of the class III homeo domain-leucine zipper (HD-ZIP III) TF genes, including *REVOLUTA* (*REV*), *PHABULOSA* (*PHB*), *PHAVOLUTA (PHV)*, *ARABIDOPSIS THALIANA HOMEOBOX 8* (*AtHB8*) and *AtHB15* in a broad range of plant species [23,24,25]. In soybean (*Glycine max*), analyzing *cis*-elements of 26 *miR166* members showed that miR166s’ promoters contained a highly conserved binding site of the HD-ZIP III family in their promoters, implying a possible HD-ZIPIII-mediated feedback regulatory loop for *miR166* expression in soybean [18]. Similarly, *Arabidopsis* plants over-expressing the *miR166a* gene exhibited reduced transcript levels of *PHB*, *PHV* and *AtHB15*, and an altered floral and leaf morphology, including stunted growth, a disrupted floral structure, an enlarged shoot apical meristem and expanded xylem tissue, indicating that the miR166 family is important for vascular and floral development [24,26,27,28,29]. In addition, *Arabidopsis* plants over-expressing *AtHB8* enhanced cell lignification and xylem formation, while *Arabidopsis* plants over-expressing *AtHB15* repressed the accumulation of lignified tissue, indicating that AtHB15 and *AtHB8* might have antagonistic roles during vascular bundle development [30].

All these results above suggest that miR166 participated in the modulation of lignin synthesis through HD-ZIP family target genes in plants; however, until now, little is known on how VvmiR166s respond to GA signals to regulate lignin synthesis, and there are no reports on the roles of miR166s in the modulation of seed-coat lignin synthesis of GA-induced seedless grape berries. For this, we isolated and determined the precise sequences of VvmiR166a-h, identified their target HD-Zip III TFs (*VvREV*, *VvHB15* and *VvHOX32*) in grape cv. ‘*Rosario Bianco*’, determined the complementary degree and cleavage role of diverse VvmiR166s-mediated *VvREV*, *VvHB15* and *VvHOX32* target genes in grape berries, and analyzed the promoters’ GA-responsive *cis*-elements of both *VvMIR166s* and *VvREV*, *VvHB15* and *VvHOX32* target genes. Subsequently, we further examined the spatiotemporal expression profiles of *VvmiR166*s, their *VvREV*, *VvHB15* and *VvHOX32* target genes in the flesh, peel and seed of the grape berries at different flowering stages, and their responsive modes to GA at corresponding tissues. And we validated their molecular functions in the modulation of lignin synthesis through the transient over-expressions of *VvMIR166s* and their target genes in *Tobacco*. Our findings present novel insights into the roles of the VvmiR166 family underlying the GA signaling repressing seed-stone lignin synthesis that promotes the seedless process of berries in grapes, which are important for the development of high-quality seedless grape berries.

## 2. Results

### 2.1. Morphological and Physiological Changes during GA-Induced Grape Seedless Berry Development

To investigate the effects of GA on grape berry and seed development, the morphological and physiological variations of an elite grape cv. ‘*Rosario Bianco*’ were observed and examined. GA treatment can conspicuously induce seedless berries with about a 98.26% seedless rate, while GA-untreated control grapes have a seedless rate of only 0.31% (Figure 1A,D). Moreover, GA application significantly increased the single berry weight of grape (Figure 1B). In addition, GA distinctly promoted the fruit shape index of grape berries, which increased from 1.19 to 1.68 along with the development of grape berries (Figure 1C). The length of the berry brush became longer, but its diameter became thinner compared with GA-untreated control plants (Figure 1A). These morphological changes reflected that some physiological and molecular responses to GA might be involved in the modulation of grape berry and seed development.

### 2.2. Identification of VvmiR166s in Grape cv. ‘Rosario Bianco’ through miR-RACE and Comparison with Their Homologous Sequences in miRBase

Our previous studies found that the VvmiR166 family could respond to exogenous GA in grape berries using small-RNA deep sequencing [15]. Here, we further cloned and verified the precise sequences of VvmiR166a/b/c/d/e/f/g/h in grape cv. ‘*Rosario Bianco*’ via miR-RACE technology. The results demonstrated that the precise sequence of all eight members of the VvmiR166 family in grapes was 21 nt in length, of which VvmiR166c/d/e/f/g/h had the same sequences as the homologous ones in miRBase21.0, but VvmiR166a/b were different from those in miRbase 21.0 (Figure 2A), exhibiting two more bases ‘TC’ at the 5′-/3′-ends of the corresponding VvmiRNAs compared to those in miRbase21.0 (Figure 2A). Further phylogenetic analysis of miR166s mature sequences showed that the genetic relationship of grape miR166s was closer to those in poplar and tobacco, and the genetic relationship with rice miR166 was further (Figure 2B). In contrast with the mature sequence, the precursor sequences of VvmiR166s possessed a lower similarity (Figure 2C). But all VvmiR166s precursor sequences can form a stable stem-loop structure, supporting the authenticity of these miRNAs (Figure 2D).

### 2.3. Verification of VvmiR166s-Mediated Cleavage Roles on Target Genes in ‘Rosario Bianco’ Grape Berries

The precise sequences of VvmiR166s were used as queries to search their target genes from the grape mRNA database (http://genomes.cribi.unipd.it/DATA/V2/V2/) (accessed on 20 October 2017). In total, four genes *VvHOX32*, *VvREV*, *VvReloluta-Like* and *VvHB15* were predicted as the target genes for each member of the VvmiR166 family, and all target regions of VvmiR166 family members were located in the coding sequence (CDS) of their target genes (Appendix A). *VvHOX32*, *VvREV*, *VvReloluta-like* and *VvHB15* belong to the HD-ZIP III class of the HD-Zip TF family (Appendix A). The complementary degree between VvmiR166a/b/c/d/e/f/g/h and their target genes *VvHB15*, *VvREV*, *VvReloluta-Like* and *VvHOX32* depicted that there was one base mismatch between VvmiR166a/b and all four target genes, while there were three base mismatches between VvmiR166c/d/e/f/g/h and all their target genes (Figure 3), indicating that VvmiR166a/b might have a stronger action on these four target genes than VvmiR166c/d/e/f/g/h. Interestingly, we also revealed that all these mismatched positions between VvmiR166s and their targets almost occurred at both ends of VvmiR166s sequences and hardly changed their target modes, which is in line with other reports [31,32], suggesting that the mismatch of their sequence ends might be one of the potential reasons for the conservation of miRNA-targeted modes within different plant species.

Most of the plant miRNAs have been shown to guide cleavage of their target genes to play roles in plant growth and development, signal transduction and responses to stress. To examine whether the three predicted targets for VvmiR166s can also be cleaved, our modified RLM-RACE and developed PPM-RACE procedures were used to map the cleavage sites in the predicted target genes from grapevines. All the anticipated RLM-RACE products (3′-end cleavage mRNAs) showing distinct bands on agarose gel were isolated and sequenced, the 3′-end cleavage mRNA sequences could be mapped into *VvHOX32*, *VvREV* and *VvHB15*, and thus these genes could be validated as the target genes for VvmiR166s, where their cleavage sites mainly occurred at the ninth site from the 5′-end from VvmiR166s at the binding region, similar to previous reports (Figure 3). Subsequently, PPM-RACE was further employed to verify these cleavage roles and cleavage sites; the results showed that PPM-RACE products (5′-end cleavage mRNAs) were mapped into *VvHOX32*, *VvREV* and *VvHB15* too; and their cleavage sites also mainly occurred at the ninth position. The consistency of cleavage sites of RLM-RACE and PPM-RACE further confirmed these genes to be the true target genes for VvmiR166s, which also had negative regulatory roles on the corresponding target genes.

### 2.4. Identification, Sequence Analysis and Sub-Cellular Localization of VvmiR166s’ Target Genes

Based on the predicted target mRNA sequences for VvmiR166s, we cloned and identified the sequences of *VvHOX32*, *VvREV* and *VvHB15* in the berries of grape cv. ‘*Rosario Bianco*’. *VvREV* had an open reading frame (ORF) of 2532 bp encoding a protein of 843 amino acid residues, and the ORF of *VvHOX32* was 2535 bp encoding a protein of 844 amino acid residues, while *VvHB15* possessed a 2520 bp ORF encoding a protein of 839 amino acid residues (Figure 4A). These three proteins all comprised four conserved domains, including the homeo domain (HD), leucine-zipper domain (Zip), steroidogenic-acute-regulatory-protein-related lipid transfer domain (START) and MEKHLA domain (Figure 4A).

We further compared the amino acid sequences of our VvHOX32, VvREV and VvHB15 proteins encoding genes with their homologous proteins derived from *Arabidopsis thaliana*, *Nicotiana tabacum*, *Fragaria vescas* sp. Vesca and *Prunus persica* (Figure 4B), and the phylogenetic tree was constructed using the MEGA software v.7 using a neighbor-joining algorithm (Figure 4C). The Gene Structure Display Server (GSDS, http://gsds.cbi.pku.edu.cn/index.php) (accessed on 25 October 2017) was also employed to analyze the intron, exon and up/downstream UTRs (Figure 4D). Although *VvREV*, *VvHOX32* and *VvHB15* all belong to the HD-Zip III family, their phylogenetic analysis divided them into the three subgroups. *VvHB15* had the closest relationship to *PtHB15*, while *VvHOX32* was highly similar to *NtHOX32*. However, *VvREV* was located at one small branch far from other homologous genes in other plant species (Figure 4C), indicating the evolutionary diversification of various members from the same gene family. Further analysis of their sequence structures showed that although the extra and intron number of these three target genes are all similar, they are different in the distribution in chromosomes. Moreover, *VvREV*, *VvHOX32* and *VvHB15* possessed longer introns, and their sequence structures are looser, in contrast with their homologous genes in the other five plant species above, implying the divergence of their evolution process (Figure 4D). Subcellular localization prediction results showed that *VvHB15* was mainly distributed in the nucleus, and *VvHOX32* and *VvREV* were mainly distributed in the nucleus and cytoplasm (Figure 5A). Transient expression in tobacco leaf epidermal cells showed that the fluorescence of *35S-VvHB15*::GFP-transformed tobacco leaf cells was only detected in the nucleus, *35S-VvHOX32*::GFP was only detected in the cytoplasm and *35S-VvREV*::GFP was localized in both the cytoplasm and the nucleus (Figure 5B).

### 2.5. Potential Function Analysis on Promoters’ cis-Elements of VvMIR166s and Their Target Genes

The promoter plays key roles in the initiation of the transcription process, and its sequence contained some essential functional components (*cis*-elements), indicating the potential functions of genes. Here, we analyzed the motif elements of *VvMIR166s* (VvmiR166s precursor genes) and *VvHB15*, *VvREV*, *VvREV-like* and *VvHOX32*. To conveniently depict the traits of these gene promoters, their *cis*-elements were classified into five different types, such as light-related elements, hormone-related elements, stress-related elements, tissue-specific elements and circadian control. As shown in Figure 6A, the *VvMIR166s* almost has five kinds of *cis*-elements except for *MIR166e* and *f* without circadian control, and among these five *cis*-elements, the number of light-related elements is the largest, accounting for more than 1/3 of the total element number, which might derive from the fact that the light is necessary for photosynthesis in all green plants. This is followed by the number of stress-related elements, hormone-related elements and tissue-specific elements. Contrarily, all four target genes for VvmiR166s have similar motif elements, of which *VvREV-like* and *VvHOX32* possessed the 5 types of *cis*-elements above, while *VvREV* and *VvHB15* had 4 out of the 5 kinds of *cis*-elements above except for circadian control (Figure 6B).

Our earlier work revealed that the VvmiR166 family could conspicuously respond to GA [15], and thus, we further analyzed the *cis*-element-related hormones for the promoters of *VvMIR166s*. As shown in Figure 6C, the promoters of *VvMIR166s* almost consist of *cis*-elements responsive to hormones like GA, ABA, Ethylene, MeJA and SA, of which the promoters of almost all *VvMIR166s* contained *cis*-elements responsive to GA except for the *VvMIR166f* promoter with only *cis*-elements related to Ethylene. This is followed by that responsive to salicylic acid being revealed at the promoters of *VvMIR166a*, *b*, *c*, *d* and *h*, and the motif responsive to MeJA and ABA, but all had no motif responsive to Auxin. These results suggested that the VvmiR166 family might be involved in the modulation of multi-hormones on grapes, of which they might play roles mainly through responding to GA and salicylic acid signals during grape berry development. On the other hand, we also observed that the diverse members of the VvmiR166 family might possess a redundancy and complementarity of the response to hormones, but they also exhibited diverse traits in the types and numbers of their *cis*-elements.

### 2.6. Spatiotemporal Expression Patterns of VvmiR166s and Their Targets during Diverse Grape Berry Tissue Development

Expression profiles of miRNAs in various tissues not only confirm the existence of the miRNAs in the organisms but also provide clues about their functions. To recognize the potential functions of VvmiR166s, qRT-PCR analysis was employed to detect the expression levels of *VvMIR166s* genes and their targets (all primers in Appendix A) in the various tissues of grape berries. The results indicated that VvmiR166s was expressed ubiquitously in three types of grape berry tissues (peel, flesh and seed), showing some tissue and/or stage-specific patterns. From Figure 7A, we revealed that almost all VvmiR166s possessed one typical characteristic, which had the lowest expression in the grape stone-hardening stage (46th DAF period), while it had a ubiquitous expression in other stages. Especially, VvmiR166d had high expressions in the seeds of the remaining stages except for the stone-hardening period (46td DAF period), followed by VvmiR166a, VvmiR166g and VvmiR166h, indicating that these members might be involved in the modulation of the development of the special stages; also, another mode of VvmiR166b, c and e expression is that their expressions in seeds and peels exhibited the ‘V’ mode at the whole level with the development of berries from the 32nd DAF period to the 80th DAF period, of which they had the lowest ebbs in the corresponding tissues at the 46th DAF period, suggesting that their regulatory roles might possess a dynamic variation during grape berry development; unlike the former two modes, VvmiR166f had high expression levels only in the seed and flesh at the 32nd DAF period, while it exhibited the low expression levels at the other stages, indicating that it might mainly be involved in the modulation of the special stage of grape berry development.

To further gain insight into the roles of VvmiR166s in grape berries, the expression levels of their target genes were analyzed. As shown in Figure 7B, the three target genes of *VvREV*, *VvHB15* and *VvHox32* all exhibited the lowest expression levels in all flesh tissues at the different tissues of various-stage grape berries, of which the first two genes possessed the highest expression levels in the seed tissues at corresponding tissues, while the last one had the higher expression levels in the peel tissues at different-stages grape berries than the flesh and peel tissues. As described above, these three genes belong to HD-Zip III involved in the modulation of vascular and xylem formation, and we deduced that in this work, they might be involved in regulating vascular and xylem formation of seed stones and peels, and thus are highly expressed in the whole seeds and peels.

### 2.7. Modes of VvmiR166s and Their Target mRNAs Responding to GA during the Development of Grape Berries

To further recognize the potential roles of VvmiR166s and their targets during GA-induced grape seedless berry development, we detected the modes of their response to GA in various tissues of berries during grape berry development (Figure 8). The results showed that the VvmiR166s family showed a down-regulated trend of 32DAF-46DAF in seed and peel tissue, but showed a variety of changes in the 32DAF-46DAF of flesh tissue. The 60DAF-86DAF also showed a different trend. All these results confirmed that the VvmiR166s family possessed tissue- and stage-specific characteristics, as well as a diversification of responses to GA.

In contrast with VvmiR166s, their three target genes exhibited the differential expression modes responsive to GA in grape berries (Figure 9). Among them, two interesting cases were observed: one is that *VvHB15* was up-regulated by GA only at the 46th DAF in seeds and peels, especially in seeds, opposite to those of several members of the VvmiR166 family, indicating GA-manipulated seed development possibly by repressing VvmiR166s’ expressions to promote that of *VvHB15*; another is that all these three target genes were obviously up-regulated by GA at all these stages in fresh tissues. These findings demonstrated that these target genes possessed diverse modes responsive to GA in grape fresh tissues with those in corresponding seeds and peels, while suggesting that the VvmiR166 family might negatively modulate *VvHB15*′s expression in the special tissues of certain-stage grape berries by responding to GA. Our findings confirmed the stage/tissue-special characterization of the VvmiR166s family in response to GA involved in the modulation of grape berry development by mediating their target genes.

### 2.8. Enhancement of Negative Regulation of VvmiR166s on VvHB15 at Stone-Hardening Stage under GA-Induced Grape Seedless Berry Development

As shown in the analysis above, VvmiR166s: *VvHB15* modules may be one of the key regulators of the VvmiR166 family during GA-induced seedless berry development. To further recognize the effect of GA on the regulation mode of Vvmir166: *VvHB15*, we analyzed the effect of GA on the correlation between VvmiR166s and *VvHB15* expression in the stone-hardening stages (32DAF-46DAF; the key phase of seed development). As shown in Figure 10, in CK, the correlation coefficient of all VvmiR166s: *VvHB15* at the expression levels was ‘1’. Interestingly, exogenous GA treatment induced changes in their expression levels, which results in the corresponding correlation coefficient changing from ‘1’ to ‘−1’. These results demonstrated that GA can effectively change the regulation mode of VvmiR166s on *VvHB15*.

On the other hand, we also found that during GA-induced grape seedless berry development, the expression of several VvmiR166 members in the seeds was down-regulated at the stone-hardening stage, thereby promoting the expression of their target *VvHB15*, while previous studies showed that *VvHB15* could be involved in the modulation of lignin synthesis [30]; thus, we deduced that the VvmiR166s: *VvHB15* module might be one of reasons that GA repressed grape seed-stone development to induce grape seedless berries. This could provide a significant clue for further elucidating the molecular mechanism of GA-induced grape seedless berries in terms of the repression of seed-stone lignin.

### 2.9. Roles of VvmiR166s-VvHB15 Module in Modulation of Lignin Synthesis Process

To recognize the function of VvmiR166s and *VvHB15*, we constructed their corresponding transient over-expression pCAMBIA1302 vector (OE-*VvMIR166s/VvHB15*) (Figure 11A), and the empty pCAMBIA1302 vector was used as the control. These vectors were transiently transformed into tobacco plants using an Agrobacterium-mediated method. After a dark culture of 3 days, DNA was extracted, and specific primers were used for PCR detection. After gel electrophoresis, sequencing was performed, which was identical to the sequence of interest (Figure 11B), confirming the transient over-expression of *VvMIR166s*/*VvHB15* in tobacco plants (Figure 11C).

In view of the potential functions of the HD-ZIP III transcription factor in the modulation of lignin synthesis of mode plants [33], we further investigated the effect of the over-expression of *VvMIR166s/VvHB15* on the expression levels of genes (*NtPAL*, *NtCCR*, *NtCAD*, *NtLaccase*, *Nt4CL*, *Nt4CL*, *NtCOMT*) in the lignin synthesis pathway. First, we found that the over-expression of *VvMIR166s* in tobacco significantly decreased the expressions of *NtHB15* and *NtHB15-like* (Figure 11D), which were highly conserved with *VvHB15*, especially the completely consistent VvmiR166s-targeted regions, confirming that VvmiR166s negatively regulated the *HB15* expression level. Moreover, we also revealed that among the *VvMIR166* family, *VvMIR166h* might be one of main regulators in this process. The over-expression of *VvMIR166h* evidently up-regulated the expression of *NtPAL4*, *NtCCR1/2*, *NtCCoAMT5* and *NtCOMT1*, followed by *VvMIR166a/b* (markedly up-regulated that of *NtCAD1*) and *VvMIR166c* (up-regulated those of *NtPAL4*, *NtCAD1* and *NtCOMT1*), while the over-expression of *VvHB15* obviously down-regulated those of *NtPAL4*, *NtCCR1*, *NtCOMT1* and *NtCAD1* (Figure 11E), opposite to those in the over-expression of *VvMIR166h/a/b*, indicating that amongst the VvmiR166family, VvmiR166h/a/b might mainly modulate the lignin synthesis through repressing the *HB15* level.

### 2.10. Pathways of VvmiR166s-VvHB15 Modules in Manipulating Lignin Synthesis during GA-Induced Grape Seedless Berry Development

To recognize VvmiR166s-*VvHB15* modules in the modulation of lignin synthesis during GA-induced grape seedless berry development, we firstly detected the expression profiles of the genes related to the lignin synthesis in this process (Figure 12A). The results showed that exogenous GA obviously represses the expression levels of *Vv4CL*/*VvCOMT1*/*VvCCoAOMTs*/*VvCCR1*/*VvCCR4*/*VvCADs*/*VvLaccases*. Together with expression modes of VvmiR166s and their target *VvHB15* responsive to GA examined above in the corresponding process, we further analyzed the correlation coefficient of VvmiR166s-*VvHB15* modules and genes in the lignin synthesis pathway (Figure 12B). We found that during the GA-induced grape seedless berry process, GA might function to inhibit grape seed-stone development through mediating various VvmiR166h/a/b/c-*VvHB15* modules to repress diverse lignin synthesis genes, where those called VvmiR166h-VvHB15-*VvPAL3*/*VvCCR2*/*VvCAD1*, VvmiR166a/b-*VvHB15*-*Vv4CL/VvCAD1/VvLac7* and VvmiR166c-*VvHB15*-*VvPAL2/Vv4CL/VvCAD1/VvLac7* were the potential crucial signal pathways in this process. All these results confirmed that out of the VvmiR166 family, VvmiR166h/a/b/c might be the key regulatory factors that modulate the synthesis of lignin mainly by negatively regulating *VvHB15* in grapes, also supported by Boualem’s work, where miR166s affected the occurrence of primary roots, lateral roots and symbiotic nodules by cutting *MtHB*in *Medicago truncatula* [33].

## 3. Discussion

The miRNA genes are mainly expressed in spatially and temporally regulated manners and play a fine-tuning role primarily at the post-transcriptional level by regulating the expression of the protein-coding genes. Although lots of GA-responsive miRNAs in grapevine have been studied in our previous work [15], their functions are still unclear. In this study, we reported the potential interaction mechanism and regulatory modes of *VvmiR166s* and their target mRNAs in response to GA to modulate the development of the seedless grape berries.

Here, the four genes of *VvHB15*, *VvREV*, *VvREV-like* and *VvHOX32*, encoding a putative class HD-ZipIII protein, were predicted as the target gene of VvmiR166s in grapes, and the bioinformatics analysis demonstrated that all targeted regions of VvmiR166s on their target genes were located in the CDS regions of the corresponding targets. And VvmiR166s interacted with their target genes with the cleavage modes, which was further verified through our RLM-RACE and PPM-RACE experiments (Figure 3). These results resemble the previous reports where plant miRNAs modulated their target genes mainly through the cleavage modes [34], demonstrating that the VvmiR166s family is targeting and acting directly on *REV*, *HB15* and *HOX32* transcription factors in the HD-ZIP family, involved in modulating plant growth and development processes, like the shoot apical meristem, floral meristem embryo and vascular and xylem formation [35,36,37].

The promoter motifs of a gene could provide significant information for the prediction of gene potential functions. We observed that the promoters of all *VvmiR166*s and their target genes had the highest motif response to light (37–71%; 43–62%), as well as to stress (9–30%; 17–26%), hormones (4–24%; 5–19%) and specific tissue (4–22%; 6–20%), indicating that they might possess some potential functions in these aspects. It is possible that light, as an essential element for plant growth, plays an irreplaceable role in the growth and development of grape berries, and the motifs’ response to hormones are also important trans-acting factors: the *VvHB15* and VvmiR166s a/b/c/d/e/g/h all have GA-responsive trans-acting factors (Figure 6). By analyzing the relative expression of the target gene and *VvmiR166s* family gene (Figure 7), our study revealed that all *VvmiR166s* genes possessed one typical characteristic where they all had the lowest expression in the grapevine stone-hardening stage and the expressional value of the target gene was higher in the lignified tissue (seed and peel). It was also revealed that the gene expression of the *VvmiR166s* family had a negative correlation with the expression levels of their target genes, as well as the gene expression of tissues treated by exogenous GA (Figure 8 and Figure 9), while GA treatment down-regulated some members’ expressions of the *VvmiR166s* family, especially in seeds, and increased the expression of target genes, showing a negative correlation. We can infer that exogenous GA treatment mediates the *VvmiR166s* family and acts on the target gene to guide the induction of grape seedless berries.

These functions reported in previous studies could provide some evidence supporting the potential roles of these genes as predicted by motif analysis in this work. Based on the phylogenetic tree of homologous genes across various plant species, we could presume the characterization of the conservation and diversification of these genes and construct the potential functions of some homologous genes. In this study, these three target genes of class III HD-Zip could be further classified into diverse sub-groups, and more importantly, their evolution processes were diverse. For instance, *VvREV* and other homologous *REVs* in other plant species were categorized into one group, while *VvHB15* and homologous *AtHB15* and *AtHB8* were categorized into another group; similarly, *VvHOX32* and orthologous *NtHOX32*, *OsHOX32* and *PpHB* were categorized into another group. These findings could provide significant information for further understanding their functions based on their conservation with the homologous genes of model plants. Previous studies have shown that *REV* is involved in the regulation of embryo development, vascular tissue, polar auxin transport, plant morphology and development, while *VvREV* was genetically closer to homologous *REV* in other plant species. On the other hand, we also revealed that *VvREV* had the highest expression levels in grape seeds, signifying the potential functions in the modulation of grape seed development (it is now being researched in our other work).

Some studies have reported that over-expressing *AtHB15* repressed the accumulation of lignified tissue [30]. We revealed that GA could repress the accumulation of lignins in seed stone of grape berries through down-regulating VvmiR166s to promote the expression of *VvHB15* through transient transgenic over-expressions of *VvMIR166s* and *VvHB15* in tobacco, respectively. Both results support the view that *VvHB15* might be one important regulatory factor in the modulation of lignin accumulation in grape seed-stone tissues. Similarly, multiple reports showed the regulation of lignin synthesis by the Homeobox HD-Zip class III transcription factor family [38,39,40]. The repression of secondary cell wall formation in the pith is under the control of the class III HD-ZIP ATHB15 TF [41]. AtmiR166s-mediated cleavage of *ATHB15* mRNA regulated vascular development in *Arabidopsis* inflorescence stems, and an ectopic lignification phenotype was reported in the *athb15* mutant [42]. And over-expression of miR166 could lead to the down-regulation of target HD-ZIP IIIs transcript levels [43]. While we also found similar results here, and the over-expression of *VvMIR166s* regulated the expressions of genes in the lignin synthesis pathway by mediating the target NtHB15, by contrast, GA might inhibit the lignin synthesis, thereby inducing grape seedless berries by repressing VvmiR166s’ expressions during grape berry development, especially at seed development. All these results supported the view of VvmiR166s-mediated target genes being involved in the modulation of grape seed and stone development.

By comparing the potential pathways of the *VvMIR166*-family-mediated lignin synthesis in tobacco and grapes, it was found that VvmiR166a/b/c/h possessed conserved functions in the regulation of lignin synthesis by cracking *HB15*. However, the significant difference is that VvmiR166a/b/c/h repressed the lignin synthesis through mediating *VvHB15* to regulate various lignin synthesis gene pathways (Figure 13), indicating the diversification of VvmiR166 members modulating lignin synthesis. As shown in Figure 13, the miR166a/b-*HB15* module regulates *NtCAD1*, the miR166c-*HB15* module regulates *NtPAL4/NtCAD1/NtCOMT1* and the miR166h-*HB15* module regulates *NtPAL4/NtCCR1/NtCCR2/NtCCoAMT5/NtCOMT1*-mediated lignin synthesis in tobacco, while in grape, the miR166a/b-*HB15* module regulates *Vv4CL/VvCAD1/VvLac7*, the miR166c-*HB15* module regulates *VvPAL2/Vv4CL/VvCAD1/VvLac7* and the miR166h-*HB15* module regulates *VvPAL3/VvCCR2/VvCAD1* involved in the regulation of lignin synthesis. Our results demonstrated that miR166a/b/c/h-*HB15* modules mediated lignin synthesis by regulating various lignin synthesis gene pathways.

## 4. Materials and Methods

### 4.1. Plant Materials and GA Treatments

Based on our preliminary study, we collected experimental berry samples from an owned rooted 5-year-old grapevine cv. ‘*Rosario Bianco*’ (*Vitis vinifera* L.), grown under common field conditions at the Jiangsu Vocational College of Agriculture and Forestry, grape farm, Jurong, China, used as the plant material. According to the production experience and variety characteristics, a total of 3 grape plants were randomly selected as experimental materials (one plant for each experiment, and three repeats), 3 clusters of inflorescence for each plant were used as treatment and another 3 were used for the corresponding plant as the control. And the clusters of inflorescence in treatments were treated with 50 mg/L GA for 30 s, and those in controls were treated with water for 30 s, at 10 days before flowering. Samples were collected at the 32th, 46th, 60th and 86th days after flowering, and in each sampling, 5 berry grains for each cluster/each time were collected in each plant. After collection, part of the samples was used for measurement of the physiological index. In the super clean bench, we performed the disinfection of another of the sample berries with 75% alcohol, cut the berry grain into halves using a scalpel, isolated the seeds, separated the peel from the flesh using tweezers, froze it immediately in liquid nitrogen and stored it at −80 °C until use for further investigation.

### 4.2. Prediction of the Targets of VvmiR166s and Their Functions Using Bio-Informatics

Our identified mature sequences of VvmiR166s represented two unique sequences of VvmiR166a/b/c/d/e/f/g/h using Multiple Align (BioXM version 2.7 software). Based on the nearly complementary traits of miRNAs and their target genes in their sequences, together with the bio-informatics methods, we could employ the PsRNATarget Software online (http://plantgrn.noble.org/psRNATarget/) (accessed on 28 October 2017)to predict the target genes for VvmiR166a/b, and c/d/e/f/g/h, respectively, and obtain their interaction mode cleavage or translation, and the function of their target genes was annotated using the Grapevine Genome CRIBI Biotech website (http://genomes.cribi.unipd.it/) (accessed on 29 October 2017).

### 4.3. Motif Analysis of the Promoters from VvmiR166s and VvHOX32, VvREV and VvHB15

From the grape genos cope database, we obtained the promoters (approximately 1500 bp upstream of genes) of the VvmiR166 precursor and *VvHOX32*, *VvREV* and *VvHB15*, and we employed the Plantcare (https://bioinformatics.psb.ugent.be/webtools/plantcare) (accessed on 5 November 2017)to predict the motif elements of the promoters of these genes. We further analyzed the potential functions of their motif elements and compared the conservation of motif elements between those from the promoters of VvmiR166s’ precursors and *VvHOX32*, *VvREV* and *VvHB15*.

### 4.4. Construction of Phylogenetic Tree

The software MEGA version 7.0 and Clustal version X2 were employed to conduct the phylogenetic analysis. Homologs of *VvHOX32*, *VvREV/VvREV-like* and *VvHB15* were identified through a blast search of the NCBI databases (https://blast.ncbi.nlm.nih.gov/Blast.cgi) (accessed on 10 November 2017) using nucleotide and amino acid sequences of these four VvHOX32, VvREV/VvREV-like and VvHB15. After the multiple alignments were performed using Clustal version X2 in this software, the unrooted phylogenetic trees were constructed using Neighbor-Joining (NJ), and the bootstrap test was carried out with 1000 iterations.

### 4.5. RNA Extraction and cDNA Synthesis

Total RNA was isolated from 200 mg of the grapevine tissues mentioned above using the modified CTAB method22. Low-molecular-weight RNA (LMW RNA) and high-molecular-weight RNA (HMW RNA) were separated using 4 M LiCl. The concentration of RNA was measured using a UV-1800 spectrophotometer and visually ascertained in a 1.0% agarose gel. LMW RNA was polyadenylated at 37 °C for 60 min in a 50 μL reaction mixture with 1.5 μg of total RNA, 1 mM ATP, 2.5 mM MnCl_2_ and 4 U poly(A) polymerase (Ambion, Austin, TX). Poly (A)-tailed small RNA was recovered through phenol/chloroform extraction and ethanol precipitation. 5′-adapters (5′-CGACTGGAGCACGAGGACACTGACATGGACTGAAGGAGTAGAAA-3′) were ligated to the poly (A)-tailed RNA using T4 RNA ligase (Invitrogen, Carlsbad, CA, USA), and the ligation products were recovered through phenol/chloroform extraction followed by ethanol precipitation. Reverse transcription was performed using 1.5 μg of small RNA and 1 μg of (dT)30 RT primer (ATTCTAGAGGCCGAGGCGG CCGACATG-d(T)30 (A, G or C) (A, G, C, or T)) with 200 U of SuperScript III reverse transcriptase (Invitrogen, Carlsbad, CA, USA). Poly (A)-tailed small RNA or mRNA (10 μL total volumes) was incubated with 1 μL of (dT)30 RT primer and 1 μL dNTP mix (10 mM each) at 65 °C for 5 min. The reactions were chilled in ice for at least 2 min, the remaining reagents (5 × buffers, dithiothreitol (DTT), RNase out, SuperScript III) were added as specified in the SuperScript III manual, and the reaction was left to proceed for 60 min at 50 °C. Finally, the reverse transcriptase was inactivated through incubation for 15 min at 70 °C, and cDNA for mRNA and poly(A)-tailed small RNA was stored at −20 °C before use.

### 4.6. Expression Analysis of VvmiR166s Using qRT-PCR

The template for quantitative real-time polymerase chain reaction (qRT-PCR) was the cDNA for poly(A)-tailed small RNA mentioned above. To amplify the VvmiR166a/b, c/d/e/f/g/h from the reverse-transcribed cDNAs, qRT-PCR was performed using SYBR Premix Ex TapTM kit (Takara, Dalian, China) and the Light Cycler^®^480 II (Basel, Roche). PCR cycling conditions consisted of an initial denaturation step at 95 °C for 30 s, followed by 40 cycles at 60 °C for 20 s and 95 °C for 5 s. Relative expression level was calculated using the formula 2^−ΔΔCT^ = normalized expression ratio. Each PCR assay was carried out using three biological replicates, and each replicate corresponded to three technological repeats of separate experiments.

### 4.7. Expression Analysis of VvHOX32, VvREV and VvHB15

The expression of *VvHOX32*, *VvREV* and *VvHB15* was assayed using *qRT-PCR* as previously described. The reverse transcription product was amplified using gene-specific primers that overlapped the known or predicted cleavage site. Reactions were performed in triplicate on the Light Cycler^®^480 II (Roche, Switzerland). The UBI gene was used as a reference gene in the qRT-PCR detection of mRNAs. Relative expression level was calculated using the formula 2^−ΔΔCT^ = normalized expression ratio. Each PCR assay was carried out in three biological replicates, and each replicate corresponded to three technological repeats of separate experiments.

### 4.8. Mapping mRNA Cleavage Sites with RLM-RACE and PPM-RACE

To map miRNA-mediated cleavage products, we developed and performed an integrated method comprising the following steps: HMW RNAs were polyadenylated at 37 °C for 60 min in a 50 μL reaction mixture with 5 μg of HMW RNAs, 1 mM ATP, 2.5 mM MnCl_2_, 5 mM 5 × buffer and 8 U poly(A) polymerase (Ambion, Austin, TX, USA), and HMW RNAs were ligated to a 5′ adapter (5′-CGACUGGAGCACGAGGACACUGACAUGGACUGAAGGAGUAGAAA-3′) using T4 RNA ligase (Invitrogen, Carlsbad, CA, USA). Then, poly (A)-tailed HMW RNA and adapter-ligated HMW RNA were recovered through phenol/chloroform extraction followed by ethanol precipitation. Reverse transcription was performed using 5 μg of poly (A)-tailed HMW RNA and adapter-ligated HMW RNA, as well as 1 μg of (dT)30 RT primer (ATTCTAGAGGCCGAGGCGGCCGACATG-d(T)30 (A, G, or C) (A, G, C, or T)) with 200 U of SuperScript III reverse transcriptase (Invitrogen, Carlsbad, CA, USA). Poly (A)-tailed HMW RNA and adapter-ligated HMW RNA (10 μL total volume) were, respectively, incubated with 1 μL of (dT) 30 RT primer and 1 μl dNTP mix (10 mM each) at 65 °C for 5 min to remove any RNA secondary structures. The reactions were chilled in ice for at least 2 min, the remaining reagents (5 × buffers, DTT, RNase out, SuperScript III) were added as specified in the SuperScript III manual, and the reaction proceeded for 60 min at 50 °C. Finally, the reverse transcriptase was inactivated for 15 min incubation at 70 °C. After the preparation of miRNA-cleaved target mRNA libraries from various organs and tissues, we pooled similar quantities of these library samples for further PCR and qRT-PCR amplification reactions. The PCR amplifications were performed using the Gene Racer 5ʹ primer and the gene-specific primers. The amplification products were gel-purified, cloned and sequenced, and at least eight independent clones were sequenced.

### 4.9. Sub-Cellular Localization

The gene sequences of *VvHB15*, *VvREV* and *VvHOX32* were amplified using PCR using the primer in Appendix A. *VvHB15*, *VvREV* and *VvHOX32* were integrated into the pCAMBIA1302 binary vector through double digestion (*35S-VvHB15::GFP*, *35S-VvREV::GFP* and *35S-VvHOX32::GFP*). *35S-GFP* was used as a blank control. The binary vector was transiently transformed into tobacco using Agrobacterium-mediated method, and after 3 days of dark culture, sub-cellular localization was observed using laser confocal microscopy.

### 4.10. Construction of the Expression Vector and Agrobacterium-Mediated Tobacco Transient Transformation

The gene sequences of *VvMIR166s* and *VvHB15* were isolated. The primers used are described in Appendix A. To develop pCAMBIA1302-*VvMIR166s* and pCAMBIA1302-*VvHB15*constructs, 500–600 bp *VvMIR166s* and 2523 bp *VvHB15* genes were independently cloned and integrated into the pCAMBIA1302 binary vectors, respectively. A single colony of *Agrobacterium tumefaciens* EHA105 (containing the recombinant plasmid) was cultured in LB liquid medium provided with rifampin (50 μg mL^−1^) and kanamycin (50 μg mL^−1^) and cultured to OD_600_ = 0.5. The bacteria were then pelleted and re-suspended in suspension buffer (10 mM MES, 10 mM MgCl_2_, pH 5.6). The bacterial suspension was then adjusted to OD_600_ = 0.8, and 100 μmol/Lacetosyringone was added before the suspension was used for permeation and it was allowed to stand at room temperature for 4 h. The bacterial suspension was immersed through injection into the leaves of 6-week-old tobacco plants. The infiltrated seedlings were then moved back to the environmental chamber and placed in the dark for 3 days.

### 4.11. Data Analysis

Excel 2019 was used for statistical analysis of data, and Origin 2021B and Adobe Photoshop 2021 were used for chart making.

## 5. Conclusions

We studied the changes in the seedless process of grape berries of White Rosa Rio induced by exogenous GA. We identified the VvmiR166s family and its target genes *VvHB15*, *VvREV* and *VvHOX32* from grape fruits, and we verified changes in their cutting effects in grape fruits. Exogenous GA treatment induced the cutting of target genes of the VvmiR166s family in the grape stone-hardening stage (32DAF-46DAF), and it mainly enhanced the cutting of VvmiR166 on *VvHB15*. VvmiR166h/a/b may be the main factor regulating lignin synthesis by inhibiting *VvHB15*. In the process of grape fruit development, GA may inhibit lignin synthesis mainly by inhibiting the expression of lignin-related genes in the ‘VvmiR166s-*VvHB15*-Lignin synthesis-related genes’ module. Our results provide a basis for the study of the mechanism of GA induction in seedless grapes.

## Figures and Tables

**Figure 1 ijms-24-16279-f001:**
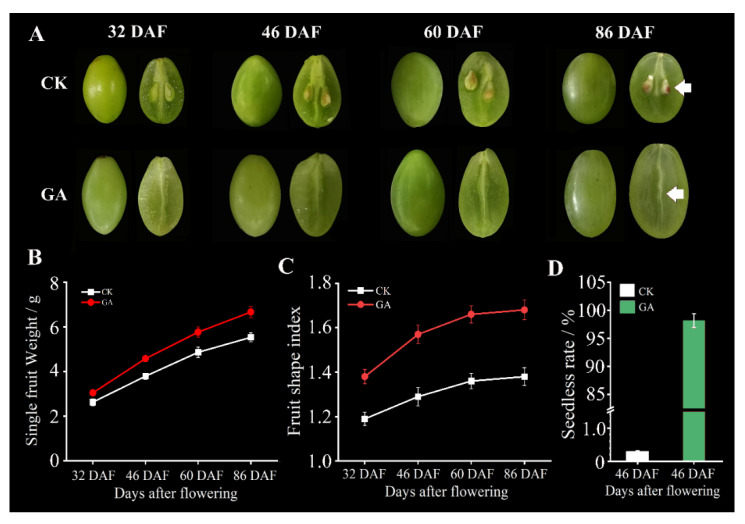
Morphological changes in berries and seeds in response to GA application. The changes in grape berries and seeds at 32 d, 46 d, 60 d and 86 d after GA treatment; the red arrow points to the berry brush (**A**). Single fruit weight changes after GA treatment (**B**). Changes in fruit shape index after GA treatment (**C**). Fruit seedless rate after 86 DAF of GA treatment (**D**).

**Figure 2 ijms-24-16279-f002:**
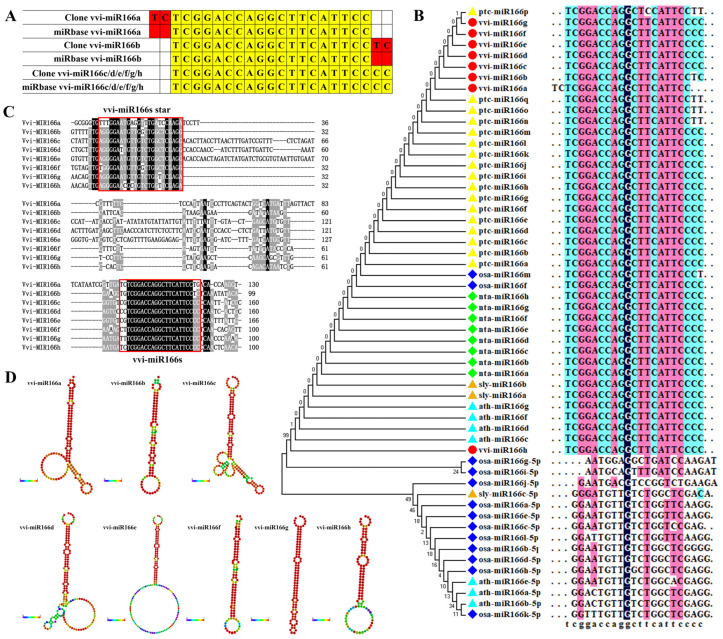
Comparison of VvmiR166s and their precursor sequences with the alignment. Comparison of VvmiR166s sequence in miRbase 21.0 and cloned VvmiR166s sequence, where red regions are different sequences and the yellow regions are the same sequences (**A**). The genetic relationship of miR166s in different species (**B**). miR166s precursor sequence alignment diagram (**C**). Stem-loop structure of miR166s precursor sequence (**D**).

**Figure 3 ijms-24-16279-f003:**
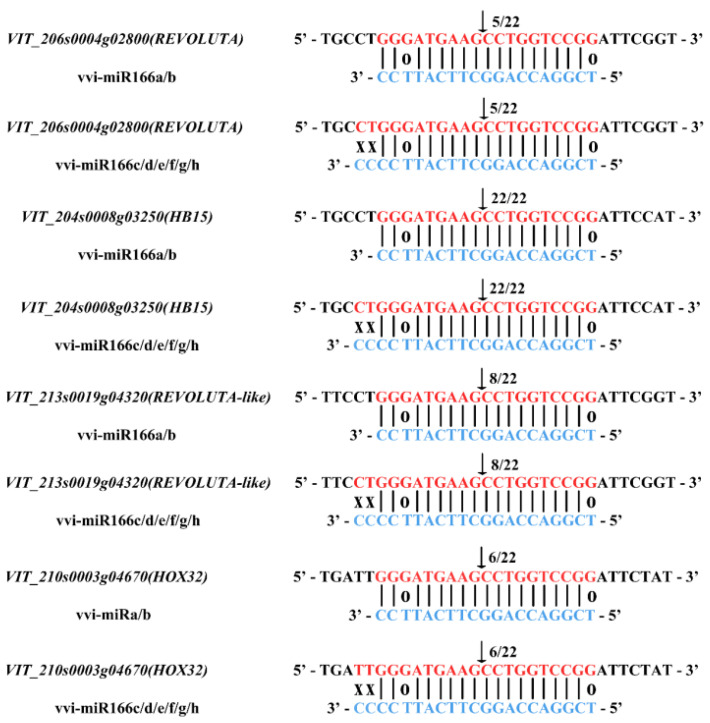
MiR166 and its target gene alignment and cutting validation. Comparison of VvmiR166s sequence in miRbase 2.1.0 and their target gene, and the mismatch bases of VvmiR166s with its target genes, where ‘X’ represents completely mismatch and ‘O’ represents the 0.5 mismatch.

**Figure 4 ijms-24-16279-f004:**
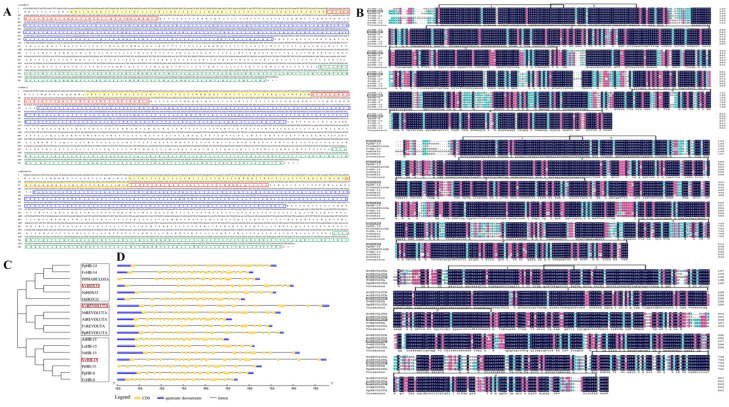
Phylogenetic tree of *HB15*, *HOX32* and *REV* across various plant species. *VvHB15*, *VvHOX32* and *VvREV* CDS sequences and their translated proteins; yellow indicates the Homeodomain (HD), red indicates the Leucine-Zipper domain (Zip), blue indicates the Steroidogenic Acute Regulatory Protein-Related Lipid Transfer domain (START), green is the MEKHLA domain (**A**). Comparison of *VvHB15*, *VvHOX32* and *VvREV* amino acid sequences in different species (1: HD; 2: Zip; 3: START; 4: MEKHLA domain) (**B**). Evolutionary tree analysis of *VvHB15*, *VvHOX32* and *VvREV* in different species (**C**). Analysis of *VvHB15*, *VvHOX32* and *VvREV* gene structure in different species (**D**).

**Figure 5 ijms-24-16279-f005:**
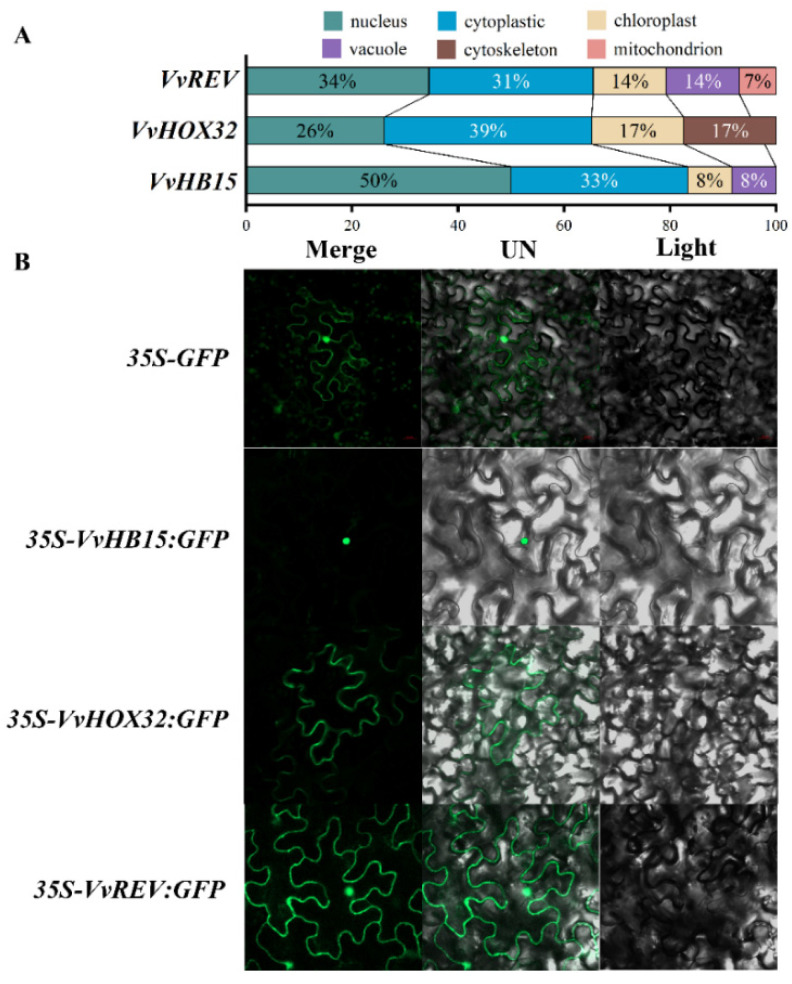
Subcellular localization of *VvHB15*, *VvHOX32* and *VvREV* in agroinfiltrated tobacco leaves. *VvHB15*, *VvHOX32* and *VvREV* subcellular localization prediction (**A**). (**B**) Validation of subcellular localization of *VvHB15*, *VvHOX32* and *VvREV* in tobacco leaves.

**Figure 6 ijms-24-16279-f006:**
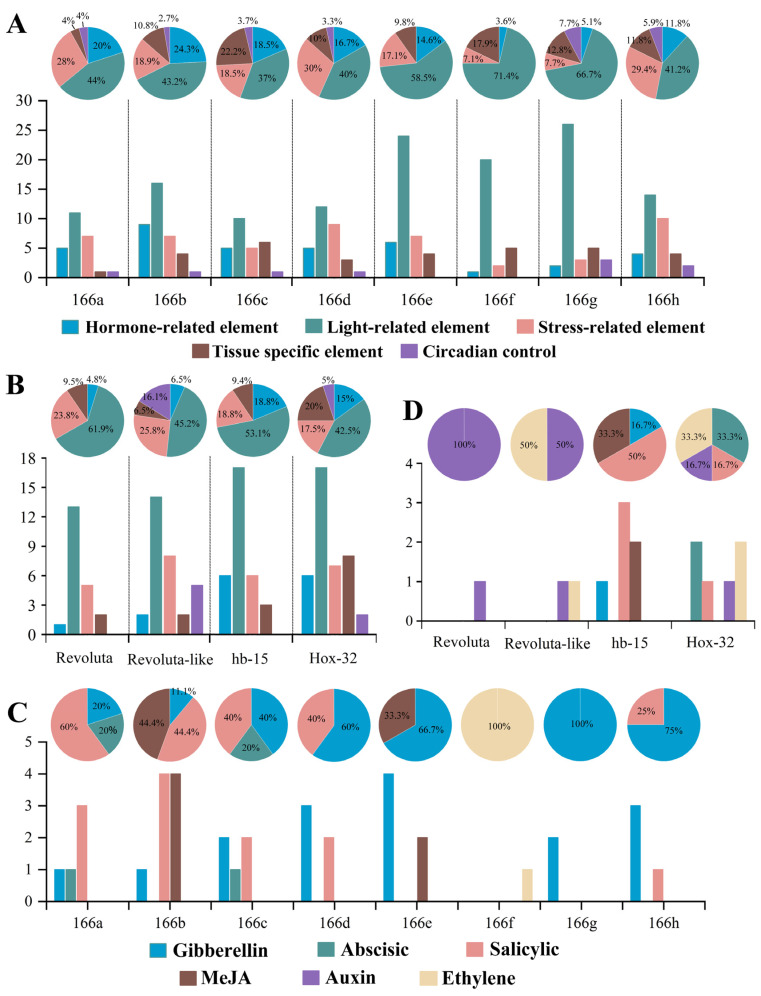
Motif analysis of promoters of *VvMIR166s* and their potential targets. *Cis*-acting element type and ratio column contained in the VvmiR166s promoter (**A**). Type and proportion of *cis*-acting elements contained in the promoter of VvmiR166s target gene (**B**). Type and proportion of hormone acting elements contained in VvmiR166s promoter (**C**). Type and proportion of hormone acting elements contained in the promoter of VvmiR166s target gene (**D**).

**Figure 7 ijms-24-16279-f007:**
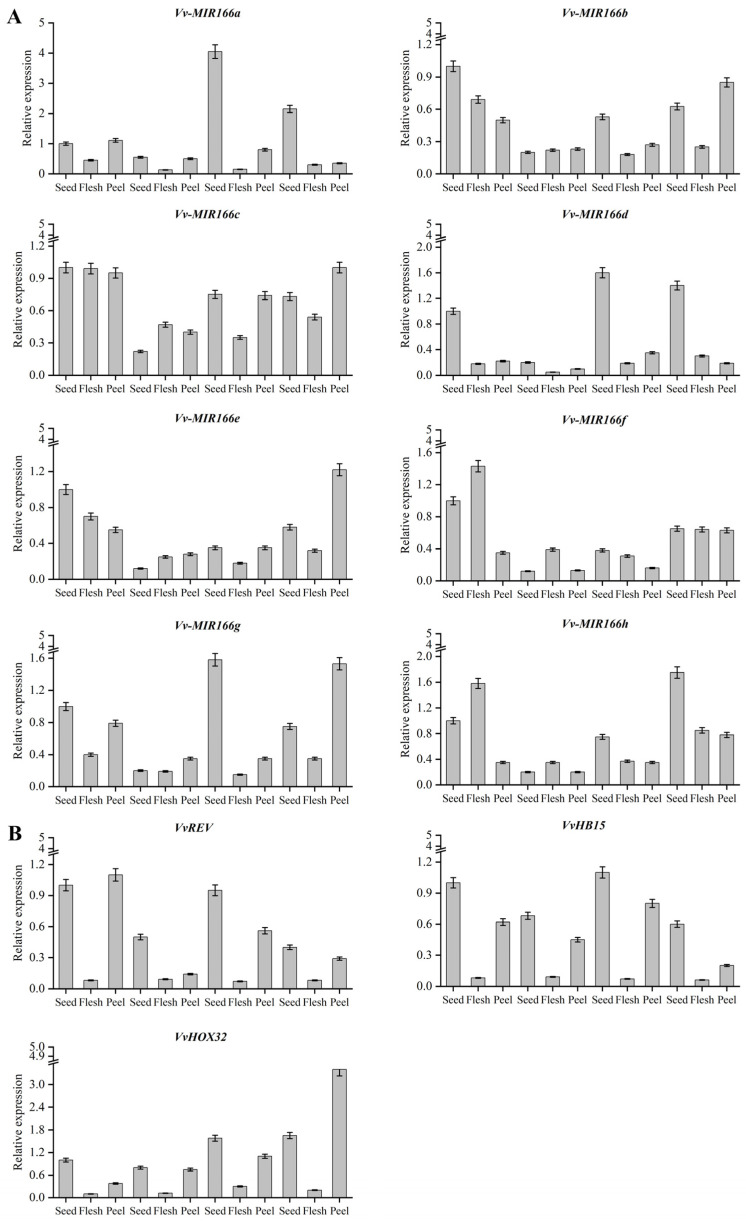
Expression profiles of *VvMIR166s* in the various tissues of diverse-stage grape berries. Expression patterns of VvMIR166s in seeds, flesh and peel (**A**). Expression patterns of *VvREVOLUTA*, *VvHB15* and *VvHOX32* in seeds, flesh and peel (**B**). 32DAF: 32nd day after flowering, 46DAF: 46th day after flowering, 60DAF: 60th day after flowering, 86DAF: 86nd day after flowering. Each reaction was repeated three times and the standard error is indicated with bars in the diagram. Each PCR assay was carried out using three biological replicates, and each replicate corresponded to three technological repeats of separate experiments.

**Figure 8 ijms-24-16279-f008:**
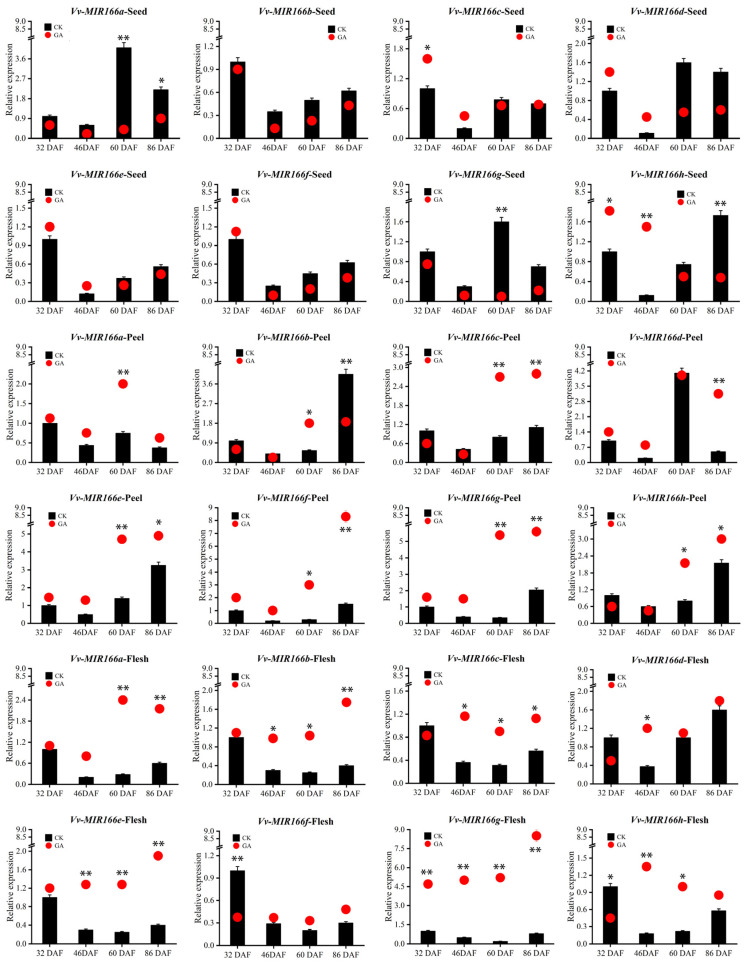
Expression modes of VvMIR166s responsive to GA in various tissues of diverse-stage grape berries. The column chart shows the relative expression of the untreated tissue, and the red dot shows the relative expression of the GA treatment tissue. Each PCR assay was carried out using three biological replicates, and each replicate corresponded to three technological repeats of separate experiments. Asterisks indicate statistically significant differences at (* *p* < 0.05; ** *p* < 0.01) as determined by Student’s *t*-test.

**Figure 9 ijms-24-16279-f009:**
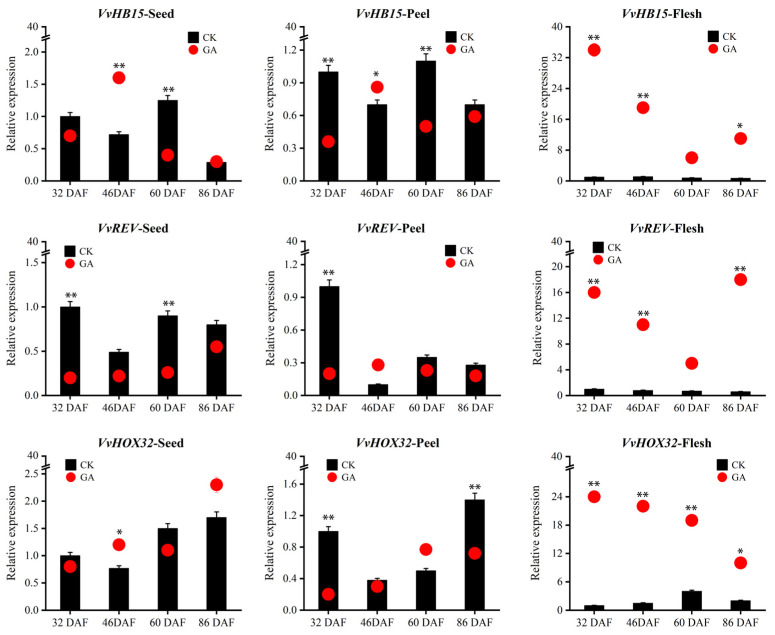
Expression modes of *VvHB15*, *VvREV* and *VvHox32* responsive to GA in various tissues of diverse-stage grape berries. The column chart shows the relative expression of the untreated tissue, and the red dot shows the relative expression of the GA treatment tissue. Each PCR assay was carried out using three biological replicates, and each replicate corresponded to three technological repeats of separate experiments. Asterisks indicate statistically significant differences at (* *p* < 0.05; ** *p* < 0.01) as determined by Student’s *t*-test.

**Figure 10 ijms-24-16279-f010:**
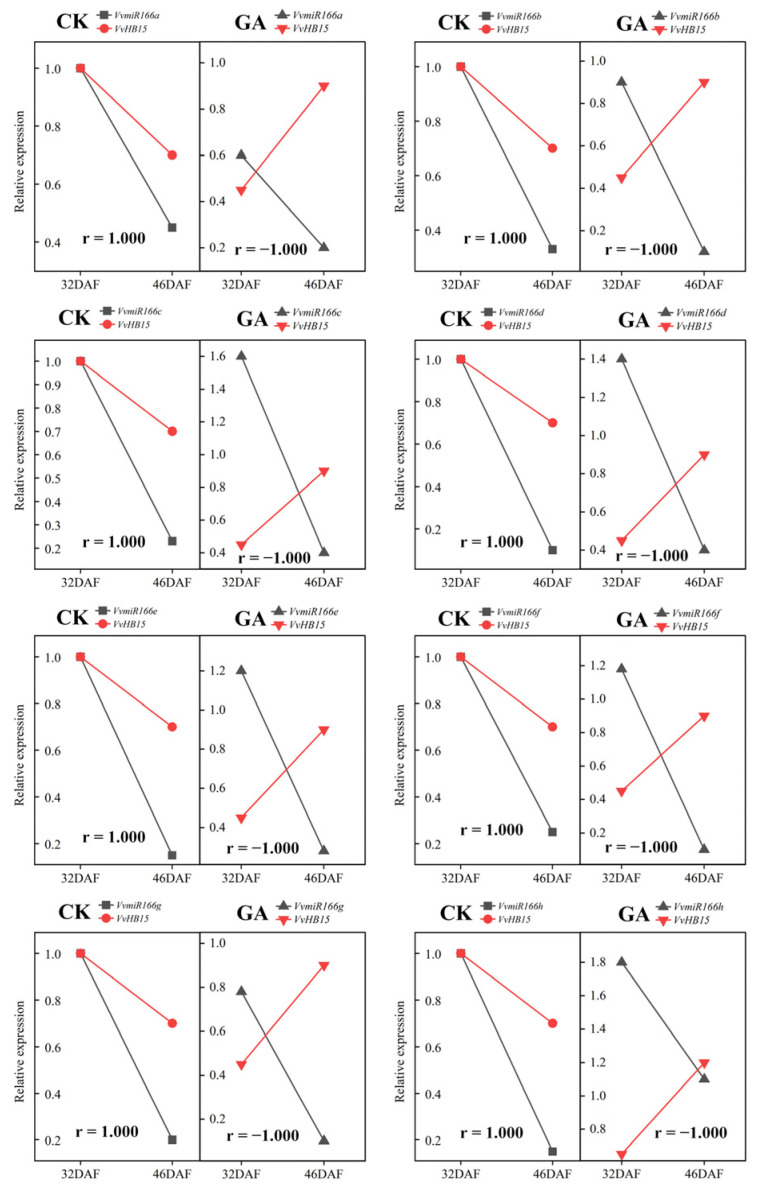
Comparison of Pearson’s correlation of VvmiR166s and *VvHB15* expression between controls and GA treatments. This chart mainly focused on the comparison of expression correlation of VvmiR166s and *VvHB15* from control and GA treatments at the stone-hardening stage (from 32DAF to 46DAF).

**Figure 11 ijms-24-16279-f011:**
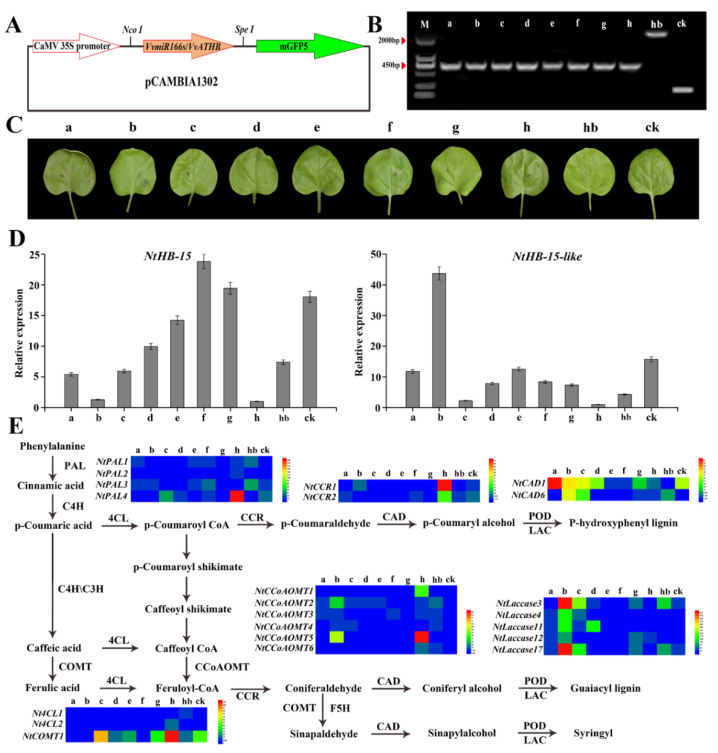
Transient expression verification of VvmiR166 family and its target gene *VvHB15* in tobacco leaves. VvmiR166a-h and *VvHB15* were integrated into the plant expression vector pCAMBIA1302 through double digestion (**A**). The obtained vector was independently transformed into tobacco plants using an Agrobacterium-mediated method. The expression vector pCAMBIA1302 was used as the control. After 3 days of dark culture, DNA was extracted, and specific primers were used for PCR detection. After gel electrophoresis, sequencing was performed, which was identical to the sequence of interest (**B**). Agrobacterium-mediated injection of tobacco leaves (**C**). After overexpressing VvmiR166a-h and *VvHB15*, the expression of *NtHB15* and *NtHB15-like* changes. Each PCR assay was carried out using three biological replicates, and each replicate corresponded to three technological repeats of separate experiments (**D**). After overexpressing VvmiR166a-h and *VvHB15*, the expression changes of genes involved in the tobacco lignin synthesis pathway were detected using real-time quantitative PCR (**E**). ‘a, b, c, d, e, f, g, h’ and ‘hb’ indicate the overexpression of VvmiR166a-h and *VvHB15*, respectively. ‘ck’ is the control.

**Figure 12 ijms-24-16279-f012:**
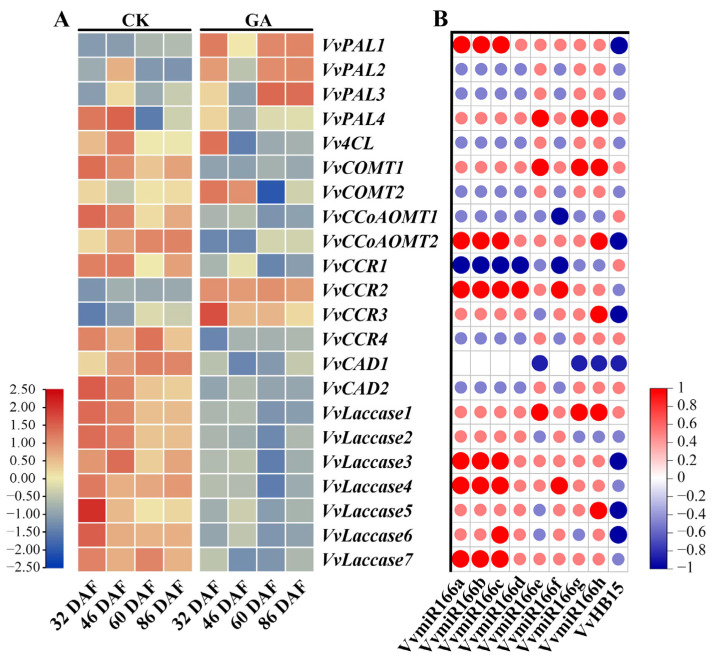
Expression of genes related to lignin synthesis in grape seeds and its correlation with VvMIR166s and *VvHB15* expression. Expression of genes related to lignin synthesis in grape seeds (**A**). Correlation of lignin-synthesis-related genes with VvMIR166s and VvHB15 expression (**B**).

**Figure 13 ijms-24-16279-f013:**
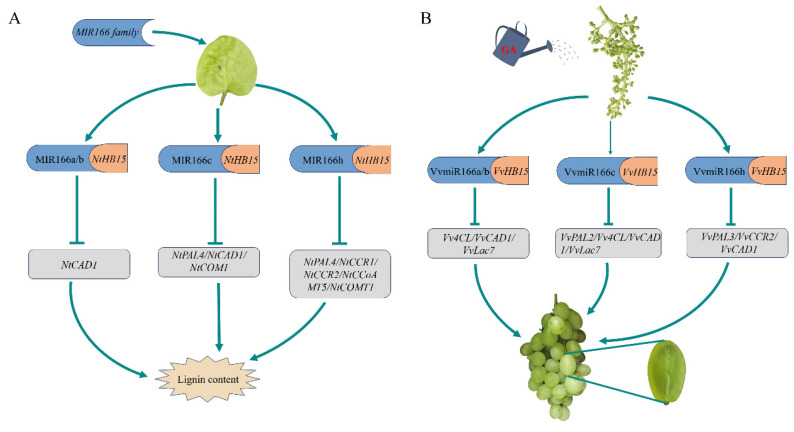
GA-VvmiR166h/b/a-*VvHB15*-genes in lignin synthesis. (**A**,**B**) Pathways of lignin synthesis regulated by miR166-HB15 modules in tobacco and grape, respectively.

## Data Availability

All data generated or analyzed during this study are included in this published article.

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
