# Peer review of "Characterization of VvmiR166s-Target Modules and Their Interaction Pathways in Modulation of Gibberellic-Acid-Induced Grape Seedless Berries"

_ijms, 2023, doi:10.3390/ijms242216279_

Round 1

Reviewer 1 Report

Comments and Suggestions for Authors      The main question is deeply insight into the role of VvmiR166s and their targets in grape 'Rosario Bianco'. Additionally, the GA-induced pathways were tested.       The topic is not so original but give some insight to the high-quality seedless grape berry. The experiments are very extensive.
   The research is primarily to learn more about the biological basis of lignin synthesis.    The methodology is correct. However, this problem could be differently resolved.            The conclusions are correctly described. The work could be developed in the next experiments. This work could have a few directions. Nevertheless, I do not see a practical application this research in Europe.     The references are appropriate.     The most are unreadable (2, 4-10). The Figures should be clear and present the results clearly. The Figures are so focused that they are difficult to read and also understand.

It's very extensive, but well done and written work.

A little correction is needed only.

The Figures 2, 4-10 are unreadable due to font mostly.

Line 198: Fragaria vescassp. - separately.

Line 425: point.

Comments on the Quality of English Language

The work is written correctly. However, the language could be more fluently partially.

Reviewer 2 Report

Comments and Suggestions for Authors

The paper is an in deep study of the effect of GA in the formation of seedless berries and the role of a miRNA and its target genes. The science is excellent and the results are sound, but there are several problems in the results presentation that must be adressed:

Figure 4: Too small and with poor resolution, but impossible to visualize. Please enlarge and enhance resolution.

Figure 7: Y axis scale is different in each pannel, so it is difficult to compare among different genes and whether the observed changes are significative as the scale changes in every pannel. Please rescale to use the same scale in every pannel, (0-5) and include a statistical analysys to check whether the observed changes are significative or not.

Figure 8, 9 and 11D: How many repeats for each bar? Please include this information in the figure legend. Also rescale to use the same scale in every pannel,  and include a statistical analysys to check whether the observed changes are significative or not.

Comments on the Quality of English Language

English must be thoroughly revised. For instance line 34-36 it is difficult to understand and there are several grammar mistakes.
